# "You always have a high sugar if you don't communicate": A qualitative secondary analysis of 'Diabetes Together' process evaluation data from couples in South Africa

**Lucy Lynch**[1]*, **Myrna van Pinxteren**[2], **Peter Delobelle**[2,3], **Naomi Levitt**[2], **Buyelwa Majikela-Dlangamandla**[2], **Kate Greenwell**[1], **Nuala McGrath**[1,4,5]

**1** School of Primary Care, Population Sciences, and Medical Education (PPM), Faculty of Medicine, University of Southampton, Southampton, United Kingdom, **2** Chronic Diseases Initiative for Africa (CDIA), Department of Medicine, University of Cape Town, South Africa, **3** Department of Public Health, Vrije Universiteit Brussels, Brussel, Belgium, **4** Department of Social Statistics and Demography, Faculty of Social Sciences, University of Southampton, Southampton, United Kingdom, **5** Africa Health Research Institute, KwaZulu-Natal, South Africa

* Lucy.lynch@soton.ac.uk

## Abstract

Type 2 Diabetes Mellitus (T2D) can negatively impact relationships which may affect health and disease management. This can be moderated by positive communication between partners. Our aims were first, to identify ways in which couples' communication and T2D impact each other. Second, we aimed to explore how improving couples' communication may enhance self-management for people living with T2D (PLWD) and their partners in South Africa. We conducted secondary data analysis of qualitative interview and focus group data from an intervention pilot study designed to improve T2D self-management – 'Diabetes Together'. 14 PLWD and their partners took part in two diabetes self-management workshops, including communication skills training, and were offered two couples counselling sessions. Inductive thematic and dyadic analysis explored how T2D and couples' communication were connected, and how participants perceived the impact of couples' communication skills training. Findings were compared to data collected from qualitative interviews with intervention-naive participants. We generated four themes linking couples' communication and T2D: tone of discussions; listening; openness; and being informed about T2D. Participants described that T2D can create negative moods and stress (both from the disease and its management) and this can make communication challenging. They reported that negative communication styles can create stress worsening T2D and making it harder to manage. Participants felt that positive communication could ease stress, enable problem-solving and support behaviour change, which may improve T2D management. Couples reported that communication skills training helped them to address communication-related issues in their relationships. This included managing negative moods, changing communication styles and developing skills to discuss, listen and collaborate on improving T2D self-management. PLWD and their partners perceived that T2D and couples' communication can affect one another positively and

**Data availability statement:** This paper includes selected excerpts from the qualitative data collected and synthesized by the authors. Releasing full transcripts would not adhere to ethics and consent practices, and therefore further information is available only upon request. For those interested in accessing the interview transcripts, access requests can be directed to risethic@soton.ac.uk.

**Funding:** NMcG, PD, MVP, BM were supported by an NIHR Research Professorship award (Ref:RP:2017-08-ST2-008) using UK aid from the UK Government to support global health research. The views expressed in this publication are those of the authors and not necessarily those of the NIHR or the UK government.

**Competing interests:** The authors have declared that no competing interests exist.

negatively. Equipping couples with effective communication skills may empower them to manage T2D more effectively.

## 1. Introduction

While diseases occur in individuals, the impacts and an individual's approach to disease management are inextricably linked to their social and interpersonal context.

As a chronic and complex condition, Type 2 Diabetes Mellitus (T2D) can negatively impact relationships which then affect health and disease management [1]. In a systematic review of 29 studies exploring the links between familial relationships and diabetes management, marriage quality was found to affect how people living with diabetes (PLWD) felt and behaved in managing their diabetes, both positively and negatively [2]. Those experiencing high marital stress showed worse blood glucose control. PLWD experienced nagging, arguing, critical comments and overprotective approaches by the partner as unsupportive. Unresolved conflicts between partners were linked with poor management of diabetes. On the other hand, PLWD who experienced higher marital satisfaction had higher diabetes-related satisfaction and felt diabetes had fewer negative impacts on their daily lives [2].

Open and supportive communication between partners can facilitate a collaborative approach to managing chronic disease. "Communal coping", a model of dyadic coping, can reduce disease-associated distress and improve outcomes [3]. In an umbrella review of how families, including intimate partners, influence T2D self-management, emotional support was identified alongside practical support in facilitating better outcomes [4]. Emotional support included understanding each other's feelings, openly communicating needs and being motivated for lifestyle changes together. In a longitudinal study of couples from the USA where one partner was experiencing a chronic condition, researchers explored how the degree of open communication and dyadic coping was connected to the relational wellbeing of the couple [5]. When couples saw the condition as a shared problem they could manage together, they discussed it more often. They used more dyadic coping strategies, which led to more satisfaction with their relationship, greater emotional closeness and higher sexual satisfaction [5].

Other research highlights the need to ensure discussions are explicit. Being clear about what the problems are, how each partner feels about them, and what solutions are desirable or preferable for each, is important to enable effective communal coping [1,6].

Couples' communication and T2D affect one another but these effects may vary according to gender and cultural background. Research in the USA found that preferences of language used to describe diabetes self-management varied between partners and PLWD, and between male and female PLWD [7,8].

There is limited research on the topic of couples' communication and T2D from low- and middle-income country settings. Notably, over half of the 29 studies in a systematic review of familial relationships and diabetes management were based in the USA with only 2 from middle- or low-income countries (Botswana and China) [2].

We recently piloted a prototype couples-focused intervention designed to improve self-management of T2D in Cape Town, South Africa. This intervention included couples' communication skills training and couples' counselling designed to enhance discussions about diabetes and enable a collaborative approach to its management [9]. We have already reported that the communication skills training was highly valued by participants and that most couples reported that their communication improved following the workshop [10].

In this paper we use inductive thematic and dyadic analysis to explore participants' views on how couples' communication and T2D impact each other and how improving couples' communication may enhance self-management for PLWD.

## 2. Methods

### 2.1  Ethics Statement

Ethical approval for the pilot study was granted by the University of Southampton (Ethics and Research Governance Online 53875) and the University of Cape Town Faculty of Health Sciences (Human Research Ethics Committee reference 031/2020). The Western Cape Department of Health gave approval for access to clinics for participant recruitment and all participants completed an informed consent process prior to participation.

### 2.2  Setting

The study took place in Cape Town, South Africa. Participants lived in low-income peri-urban settings with a predominantly black African and Coloured population. These settings face an array of challenges such as poverty, high rates of unemployment and inadequate infrastructure [11].

### 2.3  Study design

We conducted secondary analysis of qualitative data collected during pilot delivery of the prototype intervention 'Diabetes Together' [10].

The findings of this secondary data analysis are reported in line with the Standards for Reporting Qualitative Checklist (S1Appendix).

Additional information regarding the ethical, cultural, and scientific considerations specific to inclusivity in global research is included in the Supporting Information (S2Appendix).

### 2.4  Intervention

Details are provided in previous papers [9,10] but, briefly, the intervention comprised two face-to-face, half-day workshops. A summary of modules is provided in Table 1, highlighting some sessions were held with the whole group, and for some the group was split by diabetes status (PLWD and partners) or by gender.

The communication skills module aimed to improve dyadic coping with diabetes, including talking about difficult topics and resolving arguments. It was designed to help couples discuss diabetes together, communicate clearly with each other and come to a shared understanding of diabetes self-management. This module included an audio-recorded role-play, an interactive role-play for volunteer participants, a brief presentation and facilitated whole group discussion.

A core component of the communication skills module, the describe repeat method, was inspired by the speaker-listener technique [12] previously delivered in HIV-focused couples' studies [13–15]. It helps couples talk about things that matter to either one of them and which they might find difficult to raise because they are upset, annoyed or worried about how the other person might respond. The technique was illustrated with pre-recorded exchanges between partners, one of which deteriorated into an argument. The second recording acted out the describe repeat method. Volunteer participants then performed a 'live action'

**Table 1.  Diabetes Together workshops agenda.**

| Workshop 1 | Workshop 2 | |
| --- | --- | --- |
| Diabetes key facts | Substance use | |
| Eating for health | Stress management | Diabetes status groups |
| Getting active | Fears and complications | |
| Couples' communication | Sexual relationships | Gender groups |
| Goal setting | Gender roles | |

role-play. Supporting information was provided in the handouts (Fig 1) with couples invited to practice the technique as part of their homework between workshops. This approach was presented again during the second workshop, inviting feedback from participants' homework.

The initiator of the conversation (participant A on Fig 1) describes the issue to their partner (participant B). B then repeats back what they have heard in their own words. This gives an opportunity to verify what has been said and provides a pause before B responds to the content. A can then confirm B's understanding or explain further if there has been a misunderstanding. Once B has fully understood the issue, they then take on the primary speaking role and describe their response to the issue. A then has an opportunity to repeat back to B what they have heard, and the process continues until both partners feel they have been adequately heard.

A subset of couples were invited to two couples' counselling sessions after the second workshop. These aimed to build on the workshops facilitating communication and problem-solving by supporting the couples to practise the describe repeat method, to identify what they most wanted to change in diabetes self-management and to set a goal to achieve this.

## 2.5  Sampling and recruitment

Sampling and recruitment are described elsewhere [10], but in brief, PLWD were approached at primary care diabetes clinics or contacted proactively following previous involvement in diabetes management studies with the Chronic Diseases Initiative for Africa (CDIA). Convenience sampling was used to ensure a minimum number of couples were available and in recognition of the challenges of recruitment during COVID.

Eligible PLWD were aged over 18, in a relationship for at least 6 months and received care at public healthcare clinics. Eligible partners self-reported that they did not have diabetes. All participants completed an informed consent process prior to participation. Recruitment took place between 10-01-2022 and 11-02-2022.

## 2.6  Data collection

Data were collected using interviews and Focus Group Discussions (FGDs) conducted during and after pilot delivery of the prototype intervention [10]. Participants were interviewed or attended FGDs after each of the two workshops, and for couples undergoing counselling, attended individual interviews after the final counselling session. Interviews were undertaken with individual participants. FGDs involved 3 or 4 couples at a time.

Both interviews and FGDs were semi-structured using topic guides designed to capture participants' feedback on the intervention. Topic guides included questions about the communication skills module and aspects of couples' communication were also elicited through discussion and response to more general questions about how people were managing diabetes.

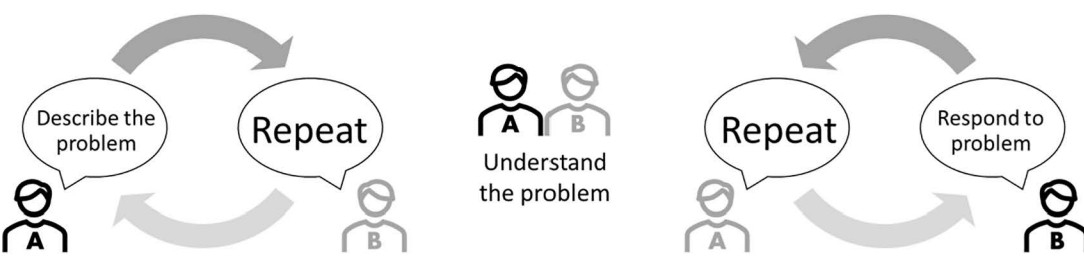

**Fig 1.  The describe repeat method.**

All interviews and FGDs were recorded and transcribed. Interpreters were present for all FGDs. Participants were given the option to conduct interviews in English or to have an interpreter (isiXhosa) present. In general, interviews and FGDs were conducted in English. At times participants responded in Afrikaans or isiXhosa and non-English language statements were translated at the time of transcription by BM and the couples' counsellor, MO. BM and MO provided quality control for each other's translations.

As previously described elsewhere, 14 couples attended the first workshop [10] on 12th February 2022. Of these, 12 completed the second workshop on 26th February 2022 and 12 contributed to at least one interview or focus group after one or both workshops. Two couples completed two couples counselling sessions (a median of 25.5 and 132.5 days after the second workshop, respectively) and a post-counselling interview (within a median of 28 days of the second counselling session). All participants were Black African (N=24) or Coloured (N=4) (see Table 2).

Table replicated from Lynch L, van Pinxteren M, Delobelle P, Levitt N, Majikela-Dlangamandla B, Greenwell K, McGrath N. 'We are in control of this thing, and we know

**Table 2. participants.**

| Study ID[a] | Gender | Age range | Language | Diabetes duration (years) | Relationship duration (years) | Interview (INT) or FGD[b] |
|---|---|---|---|---|---|---|
| 001D | Male | 60–69 | Xhosa | 4 | 36 | INT1, INT2 |
| 001P | Female | 50–59 | Xhosa | – | | |
| 002D | Male | 50–59 | Xhosa | 6 | 23 | FGD1, FGD2 |
| 002P | Female | 50–59 | Xhosa | – | | |
| 003D | Female | 50–59 | Xhosa | 6 | 3 | INT1, FGD2, INT3 |
| 003P | Male | 60–69 | Xhosa | – | | |
| 005D | Male | 50–59 | Xhosa | Unknown | 9 | None |
| 005P | Female | 50–59 | Xhosa | – | | |
| 008D | Male | 30–39 | Xhosa | 3 | 3 | None |
| 008P | Female | 20–29 | Xhosa | – | | |
| 009D | Male | 60–69 | English | 16 | 11 | INT1 |
| 009P | Female | 40–49 | English | – | | |
| 010D | Female | 50–59 | Xhosa | 20 | 12 | INT1 |
| 010P | Male | 40–49 | Xhosa | – | | |
| 011D | Female | 40–49 | Xhosa | 11 | 27 | FGD1, INT2 |
| 011P | Male | 50–59 | Xhosa | – | | |
| 012D | Male | 60–69 | Xhosa | 39 | 4 | INT1, FGD2 |
| 012P | Female | 70–79 | Xhosa | – | | |
| 013D | Male | 60–69 | Xhosa | 10 | 35 | INT3 |
| 013P | Female | 60–69 | Xhosa | – | | |
| 014D | Female | 30–39 | Xhosa | 6 | 13 | FGD1, INT2 |
| 014P | Male | 40–49 | Xhosa | – | | |
| 015D | Male | 50–59 | Xhosa | 7 | 35 | INT1, INT2 |
| 015P | Female | 50–59 | Xhosa | – | | |
| 023D | Male | 80–89 | Xhosa | 0.5 | 16 | FGD1, INT2 |
| 023P | Female | 70–79 | Xhosa | – | | |
| 024D | Female | 40–49 | Xhosa | 10 | 15 | INT1 |
| 024P | Male | 40–49 | Xhosa | – | | |

[a]D indicates PLWD, P indicates partner.

[b]Number indicates whether data collection took place after first (1) or second (2) workshop or after counselling (3).

what to do now': Pilot and process evaluation of 'Diabetes Together', a couples-focused intervention to support self-management of Type 2 Diabetes in South Africa. Glob Public Health. 2024;19(1):2386979.

Two post-doctoral researchers, one from the UK and one from SA, conducted the interviews. Both were involved in the Diabetes Together study since 2021 and are experienced in conducting individual interviews and facilitating FGDs.

## 2.7 Data analysis

LL led analysis of the pilot data for this paper. Using Braun and Clarke's 6 stages to guide thematic analysis [16], she read all interviews and FGD transcripts numerous times to become familiar with the data. She extracted all communication-related sections of text and applied inductive codes describing what the participants were saying. Once all pilot data had been coded, LL collated common topics and experiences into broader inductive themes. She analysed differences between subgroups, by diabetes status (PLWD; partners) and by gender (men; women).

Previous research inspired the approach to dyadic analysis [17]. Consideration of overlaps and contrasts in partners' reports uncovered themes about the nature of couples' relationships that would otherwise have remained invisible. The scope of dyadic analysis was limited because individual (and therefore couple) identifiers were not noted during FGD recordings. Comparison of within-couple comments was therefore only possible with data from the subset of couples who participated in post-workshop or post-counselling individual interviews. Where quotes from both partners in a couple were available for a specific code, these were compared and overlaps and contrasts between partners were noted. Communication codes described repeatedly or with emphatic statements by one partner but not mentioned by the other were also noted.

The international, multidisciplinary research team regularly discussed emerging themes, facilitating continuous reflexivity in the study. This ensured the interpretative analysis accurately reflected the contents of the data. Researchers from the University of Cape Town gave in-depth context when needed. This included exploring colloquial phrases, cultural reference points and the wider social context of the enrolled participants.

## 2.8 Data from intervention naive participants

During analysis of post-intervention data we noted that participants sometimes echoed language or messages from the communication skills module of the intervention. We did not have pre-intervention data from these participants. However, we had previously collected qualitative data about barriers and facilitators to T2D self-management from 10 couples in the same setting, recruited with the same eligibility criteria.

Interviews with intervention-naive participants explored their perceptions and experiences of living with T2D. They included questions about communication such as "Do you ever speak [to your partner] about your fears of having diabetes?" as well as exploring communication through discussion. Interviews were conducted in isiXhosa or English, recorded, translated where applicable and transcribed into English. Data from these intervention-naive participant interviews have previously been used in a secondary data analysis exploring the impact of diabetes on sexual relationships among couples, which highlighted that this topic was 'off limits' for discussion by many couples (6).

In the current study, we returned to the transcripts of intervention-naïve couples to explore if and how they talked about links between couples' communication and T2D.

Two research assistants (RAs), with previous experience of conducting qualitative interviews and coding qualitative data, independently reviewed each pair of transcripts from

intervention-naïve couples. They extracted communication-related quotes and noted concordance for each couple. The RAs discussed any differences between their data extraction and presented a final combined dataset to MvP and NMcG for discussion. LL compared these data from intervention naive participants to post-intervention data.

## 3. Results

We generated four themes describing links between couples' communication and T2D.

(1)  Tone of discussions

(2)  Listening

(3)  Openness

(4)  Being informed about T2D

Results are presented for each theme, noting any differences by diabetes status or gender, the impact of the intervention and with additional insights generated from dyadic analysis. Data from intervention-naive participants corroborated these findings.

### 3.1  Tone of discussions

Participants described challenges about the tone of their communication. There were no differences between how PLWD and partners described tone, but there were differences in how the tone of discussions were reported by gender. Men, with or without diabetes, were more often described by both themselves and their partners as angry or shouting compared to women.

Many participants reported regular short-tempered exchanges with their partners. A negative tone, commonly referred to as "angry", "shouting", "fight" and "argument", was identified as a source of stress and conflict for both PLWD and their partners.

"the way he talked to me and the way I respond, that's how conflicts start, that's how everything starts […] the tone […] shows that you're angry or that you're bitter or that you don't respect me" Female PLWD (interview 1)

"it's kind of an eye-opener because I'm someone who doesn't speak, but when I speak, it's kind of like, I kind of burst, like, 'you did that, you did that'…I think it was a learning curve for me" Male partner (interview 1)

PLWD talked about how negative tones from their partner increased their stress levels and raised their blood glucose. This was echoed by some partners.

"I said to him, 'you know you're gonna make the sugar to go high because you don't have to speak to me like that' […] What I want to say is he has that, he can't shout. We have to sit down when we have a situation at home and talk about it" Female PLWD (interview 2)

Conversely, a negative tone was sometimes ascribed by PLWD and partners to T2D – low blood sugar making PLWD feel moody or irritable.

"When you're diabetic you are becoming […] nasty or moody with everybody sometimes" Male PLWD (interview 1)

**Dyadic analysis.** Some couples gave concordant reports about the importance of a positive communication tone for effective self-management of T2D. Comments referenced the communication skills module attributing changes in actual or intended behaviour to the intervention. The statements implicitly acknowledge that prior to the intervention there was room for improvement in the tone of discussions held.

| PLWD-female | Partner-male |
|---|---|
| *"[I] never thought that communication can even affect […] your health, but [we] spoke about it [I have] learnt to communicate what makes [me] angry."* | *"So, I understand my partner. When there is a problem, I try to not speak loud to her because I know she must not be angry or maybe upset. So that's why every time I try to keep her calm […] I don't shout anymore. I don't talk loud anymore."* |
| **PLWD-male** | **Partner -female** |
| *"We practice [ways the intervention described to avoid arguing] in our home because we speak calmly. No one shouts because this raises sugar and high blood levels."* | *"To constantly be in the same accord and not having fights is key to keeping his stress levels under control. Keeping him calm is important"* |

**Impact of the intervention.** Some participants reported improvements in the tone of their conversations following the intervention.

"I learnt [from the intervention] not to shout because no one can hear if they are being shouted at." Female partner (FGD2)

"To me [the communication skills training] was very helpful because they [sic] were things that I never knew before […] just to be with my partner, and talk to my partner […] to communicate with her, but in the right direction, the right way, not always have differences. Differences can be talked over, and it's not by fighting." Male, Diabetes status unknown (FGD)

One couple reported positive changes in their communication style but had different opinions about what had needed to change in the tone of their discussions. For the female PLWD the core issue causing stress was her partner's alcohol consumption, which led him to be aggressive when discussing issues. According to her partner, though, the issue was how T2D affected his wife, making her more irritable, and through the communication skills training he had learned how to respond to this better.

| PLWD-female | Partner-male |
|---|---|
| *"Now that he has come here […] now he can talk to me when he's sober during the week, when [we can] solve things. We can sit down and say now we have a problem here and there, and we solve that thing, without him shouting."* | *"But now I learn how to speak to her, how to handle her when she [is feeling irritable]."* |

## 3.2 Listening

There were no differences in how listening was described by PLWD and partners, but there were differences in how it was described by gender.

During the men's focus group after the second workshop, all three participants described how their attitude of male dominance held prior to attending the intervention prevented them from listening adequately to their partners. This insight to gender differences in communication mirrors the women's reports under theme 1 about feeling their male partners were overly loud and domineering in tone.

"If you want happiness in your house, you have to understand what your partner says to you. Don't be 'I am a father; I can't listen to you'" Male 1, diabetes status unknown (FGD2)

"So now the communication is much better than it was before from my side now because I used to be that man who say, 'OK, I'm the one in the house, everybody should listen to me'. But now, I listen to my wife and my children also" Male 2, diabetes status unknown (FGD2)

"Things have changed a lot to an extent that there's not that cultural thing of being I am the man, I am the head of the house, it goes my way. Now [we] are partners where [we] can build together and share one another with respect equally" Male 3, diabetes status unknown (FGD2)

Overall, many participants reported how they valued being listened to and recognised the negative impact failing to listen had on how they felt and how they were able to understand or collaborate with each other, including on diabetes management.

"We always do things together but it's most important [in managing diabetes] to be...there for one another and to listen when there is any issues." Male partner (interview 1)

"we didn't understand each other so nicely [before the intervention]. Sometimes I was so fast to…say something before I listen…I didn't listen to what he saying…I just want to say what I have to say" Female partner (interview 2)

**Dyadic analysis.** Comparison of couples' reports found concordance about the value of listening. Both PLWD and their partners acknowledged that listening enables them to understand each other's point of view and recognised the impact listening has on how each partner feels. This was reported to affect both relationships and the ability to collaborate in T2D management.

| PLWD-male | Partner-female |
|---|---|
| *"Through communication you learn from your partner and your partner understands your perspective better."* | *"We also listen [...] You need to have an open mind and be able to see his perspective when we have a matter at hand"* |

| PLWD-male | Partner-female |
|---|---|
| *[interviewer: "has anything changed about the way you communicate together?"]* *"There's a lot of change […] I was very short temper and that changed […] I could not speak to my wife as a husband speak to a wife and you understand one each other and then we come closer to each other."* | *"I realize I have to listen to my husband first [...] Then he feel more at ease because he also got angry because I don't listen and I don't understand what he's trying to tell me. I just want my way and what he says […] doesn't matter and the same for him also. But since […] we come to this workshop and we learn how to speak to each other and listen […] then we know exactly what to do and how to solve the problem"* |

Some couples held different perspectives about the importance of listening within their relationship. For one couple, the female partner felt that communication was very important to enable each person to understand the others' perspective. The male PLWD felt that his wife talked too much making it difficult to pay attention to everything she said.

| PLWD-male | Partner-female |
|---|---|
| *"she said I don't pay attention to what she says. It's because she talks all the time. Non-stop talking […] so you don't know what is important […] because it's the same level she goes at all day, chatting, so it's like a monotony."* | *"the communication part is very important. So without communicating to one another, I'm not going to know what's happening in his mind, [and] he won't know what's happening in my mind"* |

For another couple, the male partner reported that, prior to the intervention, the couple would fight without resolving issues. After attending, he described being able to share and listen to problems by waiting until they were both calm to discuss issues. The female PLWD, however, reported that when trying to practice the describe repeat method, her partner became frustrated and gave up.

| PLWD-female | Partner-male |
| --- | --- |
| *"we took that section of communicating, and then talk, and then he must listen and understand and repeat what I tell him. And then he did that, and he was like, 'oh, never mind'"* | *"it taught us a lot about how to communicate with each other […] we used to fight each other and then it ended there, nothing was getting solved, [after the intervention] we were able to share our problems and those things we could sort out, so now we are just talking and then, that's when I go out, then I come back when she calms down"* |

**Impact of the intervention.**  Many participants reported how the intervention made them appreciate the value of hearing and empathising with what their partner is saying rather than interrupting or just waiting for their opportunity to speak. In some cases participants referred back to the workshop to keep each other in check. This was a benefit of the intervention being a shared experience.

"If we are in a conversation and she responds without allowing me to finish I will just say, remember the workshop?" Male, diabetes status unknown (FGD1-males)

"[The describe repeat method] will help to think what the other person feels when he just says what his problem is." Female partner, (FGD2)

## 3.3  Openness

Reports of openness varied by diabetes status but not by gender. Some PLWD reported that failing to be open created challenges for T2D self-management. For example, not sharing when they felt hungry prevented partners from providing snacks, leading to low blood sugar levels. Many partners equally wanted openness in their relationship to understand how diabetes was affecting their partner and how they could support self-management effectively.

"I accepted this cycle [of not asking for/receiving help], I started this by not allowing people in to help me. Like if I cannot be able to inject myself, I am in trouble. I know how to take my medication, that's my attitude but if I needed their help they couldn't help because of my negativity" Male PLWD (interview 1)

"He also did not want to accept his condition, and did not want to talk to me about his diabetes. He was even taking tablets I did not know what they were for, until I went to the doctor who told me what his pills were for. That's how I learnt he was diabetic. Even if he was HIV, I would not divorce him, I just want to be aware what I am dealing with" Female partner (FGD2-females)

**Dyadic analysis.**  Dyadic analysis found that openness was generally reported to elicit warm responses from partners (emotional support) and enabled PLWD to have their needs met by caring partners (practical support).

| PLWD-male | Partner-female |
|---|---|
| *"I used to be someone just speaking and saying one word and answer as you, but now I'm able to communicate positively with the whole family and not in one word, in full sentence in which I'm saying everything"* | *"They change us because I knew it was difficult to talk when you came from somewhere […] about each other, anyone or everyone look at the TV and then you go to bed. But now there is a communication"* |
| *"[at the workshop] I found that if sugar is low you can get anything to lift your sugar levels. It gives you time to explain yourself that hey, I'm […] hungry. You know, maybe someone was doing something. [...] And now I can just say 'guys I am hungry' and someone will make food"* | *"it's easier to have fruit before I give him food. Maybe sometimes I'm painting the house and he says I am hungry, then I give him fruit and it helps him"* |
| **PLWD-male** | **Partner-female** |
| *"My partner reminds to do a lot of things in relation to my diabetes management and the workshop stressed the importance of having good open communication."* | *"he is forgetful, even with checking his sugar he forgets sometimes and I remind him."* |

**Intervention impact.** Several participants reported that the intervention helped them to learn how and when to raise issues enabling partners to understand each other, feel closer and cooperate more effectively, including on self-management.

"I have also learnt [...] to not keep things inside, now I talk about what upsets me. I am not good at arguing so I learnt a lot about how to speak and when to speak" Male, diabetes status unknown (FGD1-males)

"since we have been here, she's not the same as before. I'm very open to my partner, share a lot of things. I don't keep things to myself. If there's something that I feel like is not okay with me, she must also know" Male, diabetes status unknown (FGD2-males)

"they have used that technique. [For] the partner to say what is bothering her and the partner to repeat […] what she's saying, so they can understand each other, what's the problem. It's useful because they [sic] keep the people together every time happy to speak to each other […] to understand each other" Female PLWD and male partner via interpreter (post-counselling interview)

Some participants developed the confidence to be more open with their partner by spending time with others like themselves, in groups split by gender (men; women) or diabetes status (PLWD; partners). Most participants reported that splitting into groups without their partner gave them freedom to explore difficult topics and share their own and/or listen to others' experiences.

014D-female: "I managed to speak out the way I'm feeling because he wasn't there. So, I'd like to express my how I feel…Then it was also fine that we seat all the women, only the women there, without men there, so it was good" Female PLWD (when?)

"[splitting into diabetes status groups] was great, because we learned a lot of things from one another, different things. We learnt more things, good things." Male partner (when?)

"I think he was also able to be more open minded and learn from his peers about how to be better, and me too, I also understand that our sexual relationship is affected by his diabetic condition." Female partner (FGD2)

Delivering the sexual relationships module to single gender groups ensured both partners were informed about how T2D could affect sexual health. The communication skills module helped partners to be more open about how these changes to their sex life made them feel.

Participants reported that this openness reduced conflict and enabled couples to agree a joint approach to managing the impact of diabetes on their sexual relationship.

> "For me the workshop when it came to gender roles and sex, it was useful it helped me to be able to approach my partner and tell him how I feel. And he has become less pushy about sex, and this was a result of the workshop." Female PLWD (FGD2)

> "[after the intervention] we talked about and I agreed […] if you stop [being able to have sex], then I also stop. I'm happy about it […] I can live with it you know." Female partner (interview WHEN?)

However, not all couples reported a sustained increase in openness following the intervention. One female PLWD reported that communication improved briefly after the workshop but her partner quickly reverted to being angry. The partner, however, was clear that he had changed after the intervention enabling him to support his partner with her T2D self-management.

| PLWD-female | Partner-male |
| --- | --- |
| *"I used to calm down when I talked about it with my husband, but there's no communication now. He is doing whatever he wants to do…after [the intervention] it did change, but after that, I get home after work and he's [angry]. So I ask him if we can talk, but no I can't talk to him"* | *"We used to argue a lot, but after the workshop one, after that day I just calmed myself and let it go because we need to understand each other. Sometimes it's hard for me because we go to her doctor and she says 'hey don't take any notice of my doctor' and I say 'I'm not your doctor, but I'm your husband, so if something happens to you, I'm going to think I'm responsible'"* |

## 3.4 Being informed

Lack of knowledge about how T2D could affect PLWD created challenges to self-management. There were no differences in how failing to be informed affected diabetes management by gender but there were differences for PLWD and partners.

Some participants described that lack of knowledge for PLWD led to low mood and negativity as they believed they could not manage the condition. This could result in the partner feeling pushed away.

| PLWD-male | PLWD-female |
| --- | --- |
| *"But when I bump into this diabetic thing, my attitude, everything changed, not positively changed, negatively changed"* <br> *"I've never gave anybody to learn about my sickness. I never because I classified this as a death penalty."* | *"Everything changed when he got diabetes, he feels very sad. I had to watch what I say because I know he is grumpy."* <br> *"my mother in law, she was good. She was nice. When she needed her medication, 'makoti [bride] please bring my…' and then I sit next to her. But my husband, oh, is grumpy, I leave him alone and say 'you must do yourself because you don't want me to help you, you think I will kill you […] I give you my love, but you must do it yourself because you don't know how you need me to help."* |

Other participants noted that partners' lack of knowledge about T2D could make it difficult for partners to provide effective support. For example, not understanding how T2D can impact mood could make it difficult for partners to recognise and respond when PLWD's anger might be driven by blood glucose levels.

> "With communication he is getting there but angry sometimes, does sugar make you angry? [response from another participant-female, partner "Sometimes, and you must be understanding."] "I see, thank you. I try, but the anger is too much. I don't want him shouting in front of the children" Female partner (FGD1-female)

**Dyadic analysis.** Couples' comparisons found that for some couples both partners acknowledged this lack of knowledge as a source of conflict and failure to effectively manage the diabetes.

| PLWD-female | Partner-male |
|---|---|
| *"he wasn't that supportive at first. He didn't understand what the diabetic is. I used to say to him, 'no, today I feel tired. I want to lie down, then I'm gonna wake up and make something.' Then he gonna say 'you are lazy you don't want to do like work here at home'. He gonna shout at me like I don't want to, I just want to rest"* | *"if she was feeling not well, I would not know was that thing that just happened in a minute. Yeah, both doing something and then she just go to the bed [...] So I will just say hey come here don't be lazy because I didn't understand what she said"* |

**Intervention impact.** Some participants reported that increasing their understanding of diabetes improved communication with their partner leading to better self-management. PLWD felt empowered to improve their health and more able to ask for practical help from their partners. Equally, partners felt better informed about how diabetes affected their partner and able to offer practical and emotional support.

> "I learnt a lot about how to talk to someone with diabetes and how he should talk to me. It changed many things for us." Female partner (FGD2-females)

Dyadic analysis confirmed this finding with partners reporting that increased knowledge about T2D enabled them to improve self-management through coordinated efforts.

| PLWD-male | Partner-female |
|---|---|
| *"my negativity obstructed everybody in my house. I never give people chances if I need food now, I'd rather stand up, and I would make myself food. I don't give the other people a chance to help you know, you know, but ever since we came here, now she knows, 'ok, let me stand up and improve this for this man.'"* | *"I will talk to you nicely. I will calm down when talking to him because I understand the diabetes [...] As you told me about the diabetes, I have more knowledge. I now know. Even if he is sick now I will go to him and ask what must we do [...] I'm happy for this"* |
| **PLWD-female** | **Partner-male** |
| *"It's a good thing that couples that are involved, they understand more; when I say to my husband I'm tired, why am I saying I'm tired. You understand when I say I need to be quiet. I'm not in a mood, I just need to be by myself. So they understand more now."*<br>*"It won't affect me only, it will affect him, because as you see with the diabetics, sometimes you get mood swings, sometimes you get tired, so him being there at the workshop, hearing all those things, he can understand now how it is affecting me as an individual, how also the partner can chip in, can chip in and help me live as a diabetic"* | *"So the communication is so important in our family, especially with diabetes. Sometimes she says, everything like mood swings and stuff like that, the tiredness, so even you sometimes, you [felt] like where are you going with this marriage, what is happening with this marriage?"* |

Participants reported that the sexual relationships module helped them understand how T2D could affect this part of their relationship. This helped to reduce stress and conflict.

> "[after the intervention I learned that diabetes medication] affects his sex drive [...] I was a bit disappointed [...] But afterwards I realized, well, he said he can't do it because if he stopped drinking [sic] the tablets then he will [become ill] I mustn't be selfish" Female partner (interview 2)

"[After the sexual relationships session], she understands me better than before because I used to be that person, I used to be strong you know, but now the things have changed a little bit now. In the bedroom now I'm a little bit slow, not like before, I used to be but she didn't understand but these workshops help me a lot on my side now. Now she also […] know[s] what's going on." Male PLWD (FGD2-males)

### 3.5 Secondary data analysis of data from intervention naïve participants

Intervention-naïve couples made similar points to couples interviewed after attending the intervention but used different language to describe how communication in their relationships linked to T2D. These data corroborated the 4 themes identified above, indicating couples interviewed post-intervention were not simply mirroring messages from the communication module or rehearsing what they had been taught.

A selection of relevant quotes from intervention-naive participant data for each theme is summarised in S3 Appendix.

## 4.  Discussion

This qualitative study explored the views of couples living with T2D in low-income peri-urban communities in South Africa on how communication and T2D are linked, and how couples-communication skills training affects T2D self-management. Our findings indicate couples' communication and T2D affect each other positively and negatively through tone, listening, openness, and being informed about T2D. Participants reported that improving their communication helped them to feel more understood, improved their ability to manage conflict and enabled more collaboration between PLWD and their partners in managing T2D. The findings also point to a learning curve suggesting couples develop their skills over time and that, with additional opportunities to learn, changes could be sustained. This study highlights the importance of couples' communication in T2D self-management and the potential value of communication skills training to enhance dyadic coping within a low-or-middle-income country setting.

Reflecting on our thematic analysis, our findings can be summarised as four links between couples' communication and T2D, which are consistent with existing literature about diabetes and chronic conditions more generally.

First, participants reported that poorly managed T2D can have a negative impact on communication between partners. Feeling irritable or tired due to disordered blood sugar and the stress of managing T2D, can create low moods in PLWD making it harder to talk calmly or share openly. This can leave partners feeling frustrated and unhappy. There is evidence of a relationship between diabetes and psychological disorders such as depression and anxiety that could make it difficult for PLWD to communicate easily and positively with those around them [18]. This is also consistent with work in the general population showing that higher sugar consumption over years is associated with common mental disorders [19]. These are likely to affect communication skills and style, particularly with spousal-type partners [20].

Second, negative communication styles were reported to negatively affect T2D management. This includes having a negative or aggressive tone, failing to listen and understand the other's point of view, or refusing to share or discuss issues. Such negative approaches could raise stress levels or make it harder for PLWD to manage their condition and for partners to support them. Similar issues were identified in a broad review of how family interactions could impact the management of chronic conditions, including diabetes [21]. A range of studies showed that negative patient outcomes were associated with critical, overprotective and controlling communication styles.

Thirdly, participants felt that positive communication enhances T2D management. Having open and calm discussions and listening carefully to the other's point of view can lead to a common understanding of issues, sharing of the burden and joint problem-solving. This was also found by the review of chronic conditions, showing better outcomes for patients were associated with supportive family interactions [21]. Positive communication may facilitate shared appraisal of the challenges an individual living with a chronic illness faces, enabling partners to collaborate more effectively to facilitate self-management including improving treatment adherence [22].

Finally, couples reported that communication skills training may improve communication styles and their management of T2D. The intervention provided communication skills training alongside efforts to increase knowledge about T2D and how it can be managed. Couples reported their workshop experiences led to more effective dyadic coping with the condition. This is in line with the optimised logic model for Diabetes Together [10], which incorporated part of a review of couples-based interventions for managing chronic physical illnesses [23]. The review recommended combining skills-based and relationship-focused training as both aspects were found to be effective in improving different outcomes.

There were some differences in how groups of participants described the links between T2D and communication within their relationships. Consistent with studies in the US, we found differences in language and preferences between genders of PLWD and their partners [7, 8]. We also saw some differences in communication between the genders irrespective of condition status. For example, gender affected the tone of discussions or the degree to which partners felt listened to: men were more often reported by themselves and their partners to be loud or domineering and this could be a barrier to calm discussions with both partners listening to each other. For diabetes status, PLWD were sometimes reported to struggle with being open about how their disease was affecting them or their fears about diabetes, while partners more often reported wanting and needing their partner to be more open. While lack of knowledge affected PLWD by creating fears and sadness, partners without a grounding of knowledge about how to manage diabetes were unable to offer effective support.

This study highlights the importance of communication in enabling couples living with T2D to effectively manage the condition. Communication skills training alongside education about the condition for both partners may be a valuable part of T2D self-management support.

In addition to the workshops, this intervention includes couples' counselling. These sessions were intended to build on the workshops by supporting couples to identify behaviours they would like to change and use a goal-setting approach. This involved capitalising on the new knowledge and understanding gained from the workshops, including from the greater awareness of relational aspects of diabetes management. The workshops raised issues that participants may not have previously thought about. For example, understanding how traditional gender roles might need to change following one partner's diagnosis, or building confidence to talk about the impact of T2D on sexual relationships by hearing others describe their experiences. Counsellors could facilitate greater openness between partners enabling each to understand the other's perspective, including by managing imbalance between partners in time and propensity to talk. By bringing about a shared appraisal of the issues the couple faces, the counselling sessions could facilitate practice of joint problem-solving.

Limitations of this study included limited information power [24]. As this was a secondary data analysis, we were unable to ask follow-up questions specifically in relation to communication issues that arose, which limited the depth of data collected on the topic of communication. Secondly, interviews conducted after the intervention showed participants sometimes rehearsed messages from the workshops, potentially indicating participants were

saying what they thought researchers wanted to hear. Incorporating analysis of interviews with intervention-naive couples enabled us to address this limitation and corroborate our findings. Thirdly, only two couples completed both couples' counselling sessions offered and an interview afterwards due in part to resource constraints during the COVID pandemic. Data from these interviews contributed few direct quotes about communication. Finally, the lack of participant identifiers within the FGDs prevented more extensive dyadic analysis.

## Conclusion

Our study contributes to the long-standing field of research around the role of couples in managing chronic conditions by emphasising the importance of couples' communication in self-management. We provide insights into a new community: couples from low-income settings in South Africa who are living with diabetes. Our results show the different ways communication and T2D are linked and can impact one another both positively and negatively. We found that a couples-based intervention designed to improve T2D self-management may facilitate improved dyadic coping by strengthening couples' communication skills. Future trials to evaluate this intervention should include quantitative measures of relationship-quality and diabetes-related health outcomes to assess whether self-management improves as a function of strengthened dyadic coping.

## Supporting information

**S1 Appendix:** Standards for Reporting Qualitative Research (SRQR)*.
(DOCX)

**S2 Appendix:** Inclusivity in global research.
(DOCX)

**S3 Appendix:** Quotes from intervention-naïve participants.
(DOCX)

**S4 Appendix.** Quotes from intervention-naïve participants.
(DOCX)

## Acknowledgments

Thank you to all participants in this research study. We were supported in data collection by Dr Kirsten Smith, University of Portsmouth. We would like to thank our community working group and public and patient involvement representatives for their contributions throughout the project. We would also like to thank other members of the Chronic Diseases Initiative for Africa (CDIA) team who were involved in the pilot, including: Sharon Wakefield, Chantal Stuart, Sharon Sibinda, Takalani Tshikovhi, Malibongwe Majola, Nonzuzo Mbokazi and Melinda Onverwacht, who also conducted the couples' counselling sessions. We would like to thank University of Cape Town staff and University of Southampton students who conducted transcription and/or translation including Sylvia Chaires and Morgan Blanc who joined the CHERISH programme team to review phase I data for this publication.

## Author contributions

**Conceptualization:** Lucy Lynch, Peter Delobelle, Naomi Levitt, Kate Greenwell, Nuala McGrath.

**Data curation:** Lucy Lynch, Nuala McGrath.

**Formal analysis:** Lucy Lynch, Myrna van Pinxteren, Nuala McGrath.

**Funding acquisition:** Nuala McGrath.

**Investigation:** Lucy Lynch, Nuala McGrath.

**Methodology:** Lucy Lynch, Kate Greenwell, Nuala McGrath.

**Project administration:** Myrna van Pinxteren, Nuala McGrath.

**Resources:** Nuala McGrath.

**Supervision:** Peter Delobelle, Naomi Levitt, Kate Greenwell, Nuala McGrath.

**Writing – original draft:** Lucy Lynch, Kate Greenwell, Nuala McGrath.

**Writing – review & editing:** Lucy Lynch, Myrna van Pinxteren, Peter Delobelle, Naomi Levitt, Buyelwa Majikela-Dlangamandla, Kate Greenwell, Nuala McGrath.

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
