## [Decision Letter · Decision Letter 0]

23 Oct 2024

PGPH-D-24-02126

“You always have a high sugar if you don’t communicate”: a qualitative secondary analysis of ‘Diabetes Together’ pilot data from couples in South Africa

Dear Lucy,

Thank you for submitting your manuscript to PLOS Global Public Health. After careful consideration, we feel that it has merit but does not fully meet PLOS Global Public Health’s publication criteria as it currently stands. Therefore, we invite you to submit a revised version of the manuscript that addresses the points raised during the review process.

We look forward to receiving your revised manuscript.

Kind regards,

Collins Otieno Asweto, PhD

Academic Editor

Journal Requirements:

Reviewers' comments:

Reviewer's Responses to Questions

**Comments to the Author**

1. Does this manuscript meet PLOS Global Public Health’s publication criteria ? Is the manuscript technically sound, and do the data support the conclusions? The manuscript must describe methodologically and ethically rigorous research with conclusions that are appropriately drawn based on the data presented.

Reviewer #1: Yes

Reviewer #2: Yes

2. Has the statistical analysis been performed appropriately and rigorously?

Reviewer #1: Yes

Reviewer #2: Yes

3. Have the authors made all data underlying the findings in their manuscript fully available (please refer to the Data Availability Statement at the start of the manuscript PDF file)?

Reviewer #1: Yes

Reviewer #2: Yes

4. Is the manuscript presented in an intelligible fashion and written in standard English?

Reviewer #1: Yes

Reviewer #2: No

5. Review Comments to the Author

Reviewer #1: The authors have clearly and adequately reported every detail of the study. It meets the set standard prescribed for publication. The study is a primary study whose result has not been published elsewhere. The researchers also provided the ethical consideration for holding back some raw transcripts.

Reviewer #2: Title should be more attractive and smart.

Is Food habits of T2D patients considered in this study?

Sampling method mentioned need to be more details.

Is Qualitative Data Translating process follow the standard rules?

Impressive Dyadic analysis.

Interesting findings.

6. PLOS authors have the option to publish the peer review history of their article (what does this mean? ). If published, this will include your full peer review and any attached files.

**Do you want your identity to be public for this peer review?** For information about this choice, including consent withdrawal, please see our Privacy Policy .

Reviewer #1: **Yes: ** Grace Ibor

Reviewer #2: No

---

## [Decision Letter · Decision Letter 1]

27 Jan 2025

“You always have a high sugar if you don’t communicate”: a qualitative secondary analysis of ‘Diabetes Together’ process evaluation data from couples in South Africa

PGPH-D-24-02126R1

Dear Lucy,

We are pleased to inform you that your manuscript '“You always have a high sugar if you don’t communicate”: a qualitative secondary analysis of ‘Diabetes Together’ process evaluation data from couples in South Africa' has been provisionally accepted for publication in PLOS Global Public Health.

Best regards,

Collins Otieno Asweto, PhD

Academic Editor

Reviewer's Responses to Questions

**Comments to the Author**

1. If the authors have adequately addressed your comments raised in a previous round of review and you feel that this manuscript is now acceptable for publication, you may indicate that here to bypass the “Comments to the Author” section, enter your conflict of interest statement in the “Confidential to Editor” section, and submit your "Accept" recommendation.

Reviewer #2: All comments have been addressed

Reviewer #3: All comments have been addressed

Reviewer #4: All comments have been addressed

2. Does this manuscript meet PLOS Global Public Health’s publication criteria ? Is the manuscript technically sound, and do the data support the conclusions? The manuscript must describe methodologically and ethically rigorous research with conclusions that are appropriately drawn based on the data presented.

Reviewer #2: Yes

Reviewer #3: Yes

Reviewer #4: Yes

3. Has the statistical analysis been performed appropriately and rigorously?

Reviewer #2: Yes

Reviewer #3: N/A

Reviewer #4: Yes

4. Have the authors made all data underlying the findings in their manuscript fully available (please refer to the Data Availability Statement at the start of the manuscript PDF file)?

Reviewer #2: Yes

Reviewer #3: No

Reviewer #4: Yes

5. Is the manuscript presented in an intelligible fashion and written in standard English?

Reviewer #2: Yes

Reviewer #3: Yes

Reviewer #4: Yes

6. Review Comments to the Author

Reviewer #2: (No Response)

Reviewer #3: Nil

Reviewer #4: The authors have reviewed the manuscript intelligently. Most issues raised by the reviewers have been addressed satisfactorily.

7. PLOS authors have the option to publish the peer review history of their article (what does this mean? ). If published, this will include your full peer review and any attached files.

**Do you want your identity to be public for this peer review?** For information about this choice, including consent withdrawal, please see our Privacy Policy .

Reviewer #2: **Yes: ** A K M RABIUL HASAN

Reviewer #3: No

Reviewer #4: **Yes: ** PRISCILIA UHUANMWEN IMADE
